# Rare Diseases with Periodontal Manifestations

**DOI:** 10.3390/ijerph16050867

**Published:** 2019-03-09

**Authors:** Marcel Hanisch, Thomas Hoffmann, Lauren Bohner, Lale Hanisch, Korbinian Benz, Johannes Kleinheinz, Jochen Jackowski

**Affiliations:** 1Department of Cranio-Maxillofacial Surgery, Research Unit Rare Diseases with Orofacial, Manifestations (RDOM), University Hospital Münster, Albert-Schweitzer-Campus 1, Building W 30, D-48149 Münster, Germany; lauren@usp.br (L.B.); johannes.kleinheinz@ukmuenster.de (J.K.); 2Department of Periodontology, University Hospital Carl Gustav Carus, TU Dresden, Fetscherstraße 74, 01307 Dresden, Germany; thomas.hoffmann@uniklinikum-dresden.de; 3Department of Orthodontics, School of Dentistry, Faculty of Health, Witten/Herdecke University, Alfred-Herrhausen-Strasse 45, 58455 Witten, Germany; lale.hanisch@gmail.com; 4Department of Oral Surgery and Dental Emergency Care, Faculty of Health, Witten/Herdecke University, Alfred-Herrhausen-Strasse 45, 58455 Witten, Germany; korbinian.benz@uni-wh.de (K.B.); jochen.jackowski@uni-wh.de (J.J.)

**Keywords:** rare diseases, periodontal manifestations, oral manifestations of systemic diseases, oral manifestations, classification of periodontal and peri-implant diseases and conditions

## Abstract

*Background*: The object of this paper was to provide an overview of rare diseases (RDs) with periodontal manifestations and allocate them to relevant categories. *Methods*: In ROMSE, a database for “Rare Diseases with Orofacial Involvement”, all 541 entities were analyzed with respect to manifestations of periodontal relevance. Inclusion criteria were periodontally relevant changes to the oral cavity, in accordance with the 2018 version of the Classification of Periodontal and Peri-Implant Diseases and Conditions. Rare diseases were recorded, using the methodology described, and subsequently compared with the Orphanet Classification of Rare Diseases. *Results*: A total of 76 RDs with periodontal involvement were recorded and allocated in accordance with the Classification of Periodontal and Peri-Implant Diseases and Conditions. Of the 541 RDs analyzed as having known orofacial manifestations, almost 14 percent indicated a periodontally compromised dentition. *Conclusions*: Around 14 percent of RDs with an orofacial involvement showed periodontally relevant manifestations, which present not only as a result of gingivitis and periodontitis, but also gingival hyperplasia in connection with an underlying disease. Thus, dentists play an important role in therapy and early diagnoses of underlying diseases based on periodontally relevant manifestations.

## 1. Introduction

According to the European Union (EU) definition, a disease is classified as “rare” when fewer than one in 2000 people are affected by it. At least 30 million people in the 28 member states of the EU are affected [1]. Worldwide, between 5000 and 8000 different rare diseases (RD) are known, 80 percent of which have a genetic cause [2,3]. Around four million people in the Federal Republic of Germany are affected by RDs [2]. Since 2009, there has been an increase in public awareness of RDs in the EU after the Council of the European Union called upon member states to draw up, at the appropriate level, plans and strategies for RDs [4]. In most cases, exact epidemiological data are not available as a result of incomplete registrations of RDs in national and international databases. To improve this situation, the German Ministry of Health set up a “Nationaler Aktionsplan für Menschen mit Seltenen Erkrankungen” (National Plan of Action for People with Rare Diseases) [5]. In the meantime, so-called European Reference networks for rare diseases have been set up for some RDs [6].

Rare Diseases can manifest themselves in nearly all organs and, as a result, they often have systemic effects. Their development is frequently chronic, progressive, and degenerative, and they can cause disabilities that impair the quality of life as well as life expectancy [7]. Patients who are affected, and their families, often face great difficulties regarding diagnoses and treatments [8]. Doctors often lack the time and resources, with little available information, to help them diagnose RDs accurately and, in turn, treat them properly [9]. In addition, incorrect diagnoses lead to mental stress for sufferers and their families [10]. As 15 percent of all RDs can manifest themselves orofacially [11,12], dental professionals play a major role in making diagnoses and deciding on treatment in their particular fields. This also applies to changes relating to the gingiva and periodontium [13]. However, there has always been room for improvement in both diagnoses and treatments, as well as in research and the exchange of information and experience with RDs [14].

To this day, there is no index of periodontally relevant RDs, nor any list of periodontal manifestations in RDs. Therefore, the object of this paper was to provide an overview of periodontal manifestations of RDs, as well as to allocate RDs to relevant categories, analogous to the 2018 version of the Classification of Periodontal and Peri-Implant Diseases and Conditions [15,16,17,18,19,20,21,22,23,24]

## 2. Methods

### 2.1. System Behind the Search Criteria

In ROMSE, the database for “Rare Diseases with Orofacial Involvement” [25], which went online 13 July 2017, all entities recorded were analyzed with respect to manifestations of periodontal relevance, such as gingivitis, periodontitis, gingival hyperplasia, or haemorrhagic diatheses. The database is based on the systematic literature research on orofacial manifestations, which has been in operation since 2011. To detect orofacial manifestations in rare diseases, in November 2011, work commenced on inspecting and evaluating rare disease databases such as Orphanet, (Online Mendelian Inheritance in Man (OMIM) (http://www.omim.org), and PubMed (https://www.ncbi.nlm.nih.gov/pubmed). The criterion for analysis of a rare disease and its orofacial manifestations was inclusion in the Orphanet Classification of Rare Diseases. This classification defines a ‘rare’ disease using the same definition as the EU. Thus, only rare diseases that are defined as rare in the EU were covered. The requirement of inclusion was the description of at least one dental, oral, maxillofacial, periodontal, or orthodontic symptom listed in the Orphanet, OMIM, or Pubmed databases. Medical observations in Orphanet and OMIM were analyzed for the presence of any orofacial manifestation in the respective rare disease. Our relevant literature search in Pubmed was obtained using the following enhanced keywords: “disease name” AND “cleft palate” OR “dental” OR “dysgnathia” OR “gingivitis” OR “gingival enlargement” OR “gingival hyperplasia” OR “gum bleeding” OR “hypodontia” OR “micrognathia” OR “mucositis” OR “oral health” OR “oral lesions” OR “oral and maxillofacial surgery” OR “orthodontic” OR “periodontology” OR “periodontitis” OR “prognathism” OR “stomatognathic system” OR “tooth loss” [26]. At the time of this study, 541 RDs with orofacial manifestations were listed in the database.

### 2.2. Criteria for Inclusion and Exclusion

Inclusion criteria for this publication were all periodontally relevant changes in the oral cavity, in accordance with the 2018 version of the Classification of Periodontal and Peri-Implant Diseases and Conditions [15,16,17,18,19,20,21,22,23,24]. The tongue, buccal mucosa, lips, roof of the mouth, enamel, and dentin were all disregarded. Thus, the periodontium includes the following:-Gingiva, which surrounds the tooth and parts of the alveolar bone at the level of the neck of the tooth.-Alveolar bone, which is part of the jaw bone in which the teeth are set.-Root cementum, which covers the root surface of the teeth and parts of the apical root canal wall.-Periodontal fibre, which are bundles of collagen and (partly) oxytalan fibres that reach into the root element on one side, and into the alveolar inner bone tissue on the other side.

All of the RDs recorded using the methodology described were subsequently compared again with the Orphanet Classification of Rare Diseases [27]. Only those diseases which, in line with the Orphanet Classification, were listed as RDs in accordance with the EU definition continued to be included in the list. All of the diseases were discussed among the authors on the basis of the literature available on PubMed, with a view of any periodontally relevant manifestations [28], and allocated in accordance with the 2018 version of the Classification of Periodontal and Peri-Implant Diseases and Conditions [15,16,17,18,19,20,21,22,23,24]. If no consensus was reached, the decision was made by a periodontological expert (T.H.), who is professor for “Periodontology”. We discussed together whether diseases that occur with insulin-resistant diabetes should automatically be accepted as “periodontally relevant”. We decided against it if the actual disease does not cause periodontally relevant symptoms. Thus, only diseases that are clearly associated with periodontal relevance were included.

The ethical approval for this study was obtained from the ethical review committee (Ref. no. 2017-443-f-N), Ethikkommission der Ärztekammer Westfalen-Lippe und der Westfälischen Wilhelms-Universität, Münster, Germany.

## 3. Results

A total of 76 RDs with periodontal involvement were recorded. Accordingly, of the 541 RDs analyzed as having known orofacial manifestations, 13.86 percent showed periodontally compromised dentition. The relation between the RDs and periodontal manifestations, such as possible pathological mechanisms related to the tissue alterations or affected gene and location of the disease are described in Table 1.

Table 1 summarizes 76 RDs with pathological changes in the periodontal region. Besides single entities, the specific periodontal symptoms, the pathological mechanism (if known) that is responsible for the periodontal manifestation, and the corresponding gene locus are mentioned. The list is completed by assignment of the periodontal symptoms to the 2018 version of the Classification of Periodontal and Peri-Implant Diseases and Conditions [15,16,17,18,19,20,21,22,23,24].

The pathological mechanism leading to periodontal alteration in the case of RD is marked as “unknown” in Table 1 when it is described as “unknown” or “not researched” in the literature. In 1 RD, the pathological mechanism is characterized as “not reported” according to the literature. For 6 other RDs, there are no indications in the literature regarding the pathomechanism, and in 2 RDs, the exact pathomechanism is unknown. In the case of 10 RDs, periodontal changes are considered as “not genetically determined”. In 1 RD, the gene causing the periodontal symptoms is not known.

In 21 RDs, clinical symptoms are described as “periodontitis”; in 16 RDs, as “gum bleeding, bleeding diathesis“; in 13 RDs, as “gingival enlargement“; in 7 RDs, as “gingival enlargement, fibromatosis“; in 5 RDs, as “gingivitis, periodontitis“; in 3 RDs, as “gingival enlargement, gum bleeding“; in 3 other RDs, as “gingivitis”; and in 2 RDs, as “periodontitis, gingival enlargement“. According to the literature, 1 RD each manifests as “mucosal ulcers“; “periodontal breakdown“; compromised periodontal attachment; “periodontitis, gum bleeding, bleeding diathesis"; “gingival hyperkeratosis“; “gingival candidosis“; or “gray to gray-black staining of gingiva“.

Some of the pathological mechanisms leading to periodontal involvement can be summarized in particular categories. In 17 of the RDs listed in Table 1, the periodontal alteration is related to a blood-clotting disorder. In 14 RDs, changes in the immune system are held responsible as cause for pathological processes in the periodontal region; in 3 RDs, periodontal alterations are attributed to disorders of vitamins, minerals, and trace elements metabolism; and 2 RDs are blistering disorders.

After allocation of the diseases in accordance with the Classification of Periodontal and Peri-Implant Diseases and Conditions [15,16,17,18,19,20,21,22,23,24], 39 RDs were allocated to non-plaque induced gingival diseases and conditions: genetic/developmental disorders, 15 showed manifestations that could be allocated to systemic diseases and conditions that affect the periodontal attachment apparatus: diseases associated with immunologic disorders, 4 RDs were allocated to non-plaque induced gingival diseases and conditions: neoplasms, and 3 to non-plaque induced gingival diseases and conditions: granulomatous inflammatory conditions (orofacial granulomatosis). Moreover, a total of 3 diseases were allocated to systemic diseases and conditions that affect the periodontal attachment apparatus: other disorders that may affect periodontal tissue. Three RDs could be allocated to non-plaque induced gingival diseases and conditions: endocrine, nutritional, and metabolic disease; a further 3 RDs could be allocated to systemic diseases and conditions that affect the periodontal attachment apparatus: metabolic and endocrine disorders; 3 other RDs could be allocated to systemic diseases and conditions that affect the periodontal attachment apparatus: diseases affecting connective tissues; another 3 RDs could be allocated to systemic diseases and conditions that affect the periodontal attachment apparatus: diseases affecting the oral mucosa and gingival tissue; and 1 RD could be allocated to non-plaque induced gingival diseases and conditions: gingival pigmentation.

## 4. Discussion

On the basis of the findings of this study, an association between periodontal alterations and rare diseases was observed. Periodontally relevant findings were demonstrated in almost 14 percent of the RDs in which orofacial manifestations had already been diagnosed. In another paper [29], 106 RDs with periodontally relevant manifestations were identified. However, the results provided by Müller 2017 are questionable, as some of the identified RDs could not be found in the Orphanet Classification of Rare Diseases [27]. Furthermore, certain periodontal manifestations were not verifiable, or were questionable, such as a disease involving insulin resistance or diabetes. For the cri du Chat Syndrome, it was indeed found that a Tannerella forsythia periodontal pathogenic bacterial species was more prevalent [30,31]; however, no patients were found to suffer from active periodontitis. Thus, the syndrome was not considered periodontally relevant.

It is well known that the periodontal diseases may be associated with systemic conditions, such as metabolic diseases, coagulation disorders, or drug administration. [32]. Additionally, the literature shows that a great number of rare diseases will present an implication on the periodontal tissue. The etiological components for these alterations are mostly related to the pathogenesis of the disease [33,34,35], although other factors, such as poor hygiene, may be associated with the RD [36]. In this sense, it is suggested that periodontal manifestations may be a key for the diagnosis of rare diseases.

Multisystemic diseases such as Langerhans cell histiocytosis (LCH) can simulate aggressive parodontitis. Indeed, LCH was initially incorrectly interpreted as being such a disease [37]. However, if LCH is detected and correctly classified at an early stage, systemic therapies can be immediately initiated, thereby stopping the spreading of the symptoms (Figure 1, Figure 2 and Figure 3). Changes to the gingiva, such as gingival hyperplasias, that cannot be initially classified as being a particular disease and that might be incorrectly interpreted as being medication-induced [13], are possible initial indications of the recurrence of a severe multisystemic disease such as granulomatosis with polyangiitis (Figure 4) or leukaemia (Figure 5). Serious periodontal diseases can also, at an early stage, provide differential diagnostic indications of diseases such as Ehlers–Danlos syndrome (Figure 6), which can lead to a greatly reduced quality of life, including with respect to oral hygiene [38].

Other available reports demonstrate that many oral health problems, as periodontal diseases and lower saliva flow rates, were found in subjects with neurodegenerative disease when compared with same age healthy individuals [39].

Hence, the professional must be aware of these aspects in order to diagnose periodontal disease correctly and to treat the patient accordingly.

This highlights the great importance of dental expertise in any early diagnosis. The list of RDs with periodontal manifestations in Table 1 should thus assist with the consideration of possible differential diagnoses in the event of there being a manifestation of unclear oral findings or observations that cannot be classified. Further information can then be retrieved from sources such as OMIM, Orphanet, or Pubmed that are stored in the database.

Classification systems are necessary as they provide a framework for scientific studies relating to aetiology, pathogenesis, and therapies [32]. To our knowledge, this study presents the first publication reporting RDs and their periodontal relevant manifestations with the view of facilitating the allocation of oral symptoms to a rare disease or of an oral disease to the oral symptoms as seen in clinical everyday life. Our work might thus extend the informative value of the new Classification of Periodontal and Peri-Implant Diseases and Conditions, which does not mention each of the specific rare diseases. An international network should now be developed, such as that for some RDs in the EU in the form of the European Reference Network [6], to enable the oral manifestation of RDs to be recorded in a targeted manner in the future. Dental medicine has not been included in this network to date.

### Limitations of the Study

The RDs analyzed in this study were described not by dentists, oral surgeons, or maxillofacial surgeons, but were based on cases in medical or genetic publications. The reason for this is the scant availability—or the complete lack of—dentistry publications.

Also, in the cases in which the study group made an individual assessment of the RD in question, based on the literature available, it is possible that some allocations of orofacial findings are wrong.

## 5. Conclusions

Approximately 14 percent of RDs with orofacial involvement also show manifestations in the form of gingivitis, periodontitis, and gingival hyperplasia in connection with the underlying disease. As a result, the dentist has an important role to play not only in the (accompanying) treatment, but also in any early diagnosis of the underlying disease, if he critically questions the symptoms, as described in this study, and can attribute them to a rare disease in individual cases. Also, the influence of these diseases on oral health and appropriate dental treatment to improve oral health should always be considered.

With the help of the Classification of Periodontal and Peri-Implant Diseases and Conditions, the scientific community is able to record RDs with periodontal manifestations more accurately in the future. This would mean that clinical data could be obtained more precisely to be able to assess the healthcare requirements for RDs.

In the future, there should be structured, standardized recordings of the symptoms of RDs, such as through a European alliance of university hospitals.

## Figures and Tables

**Figure 1 ijerph-16-00867-f001:**
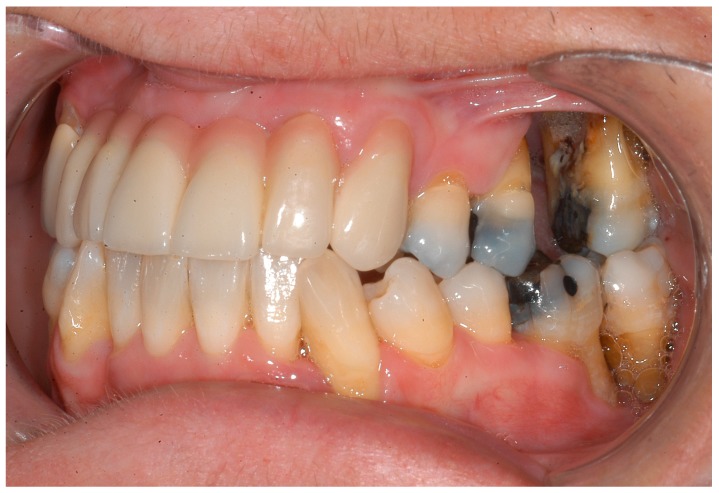
Excessive periodontal destruction in a 48-year-old women with Langerhans cell histiocytosis.

**Figure 2 ijerph-16-00867-f002:**
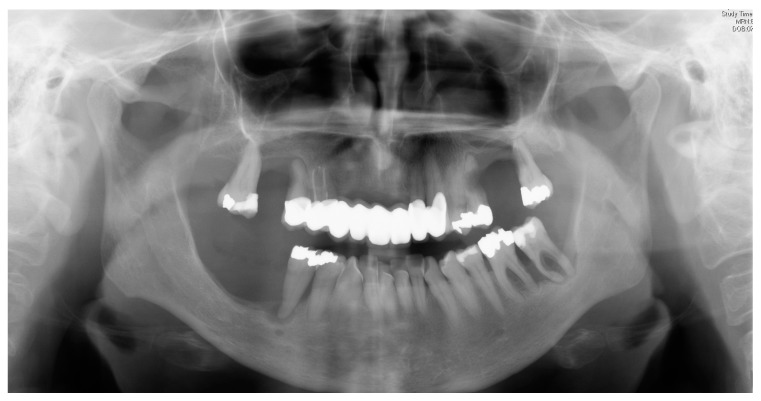
Excessive periodontal destruction in a 48-year-old women with Langerhans cell histiocytosis.

**Figure 3 ijerph-16-00867-f003:**
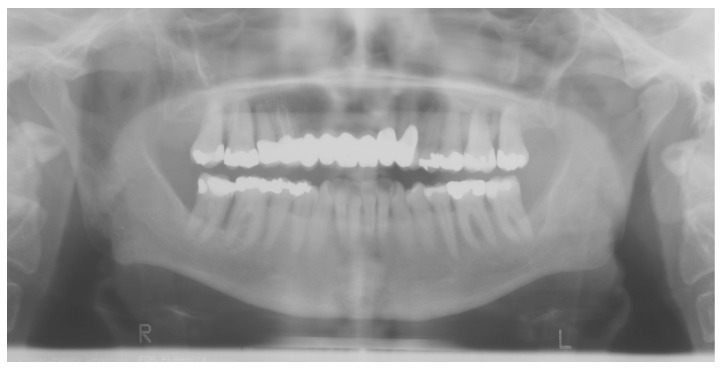
Same patient 12 months before: periodontal destruction was misdiagnosed and treated as aggressive periodontits.

**Figure 4 ijerph-16-00867-f004:**
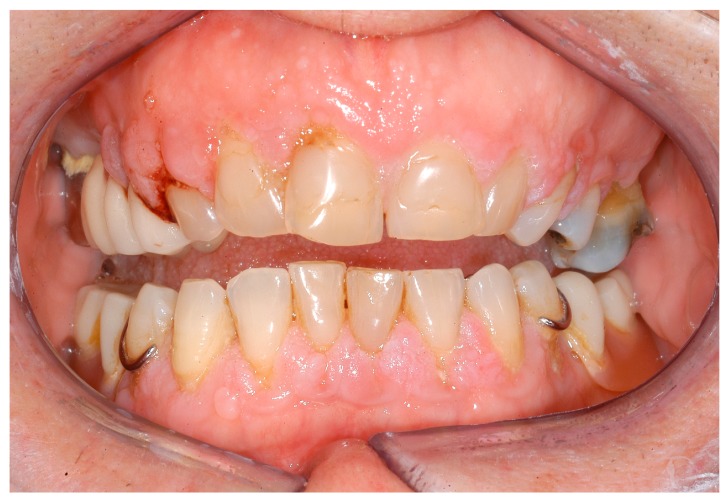
Gingival enlargement as a first sign of Granulomatosis with Polyangiitis in a 72-year-old women.

**Figure 5 ijerph-16-00867-f005:**
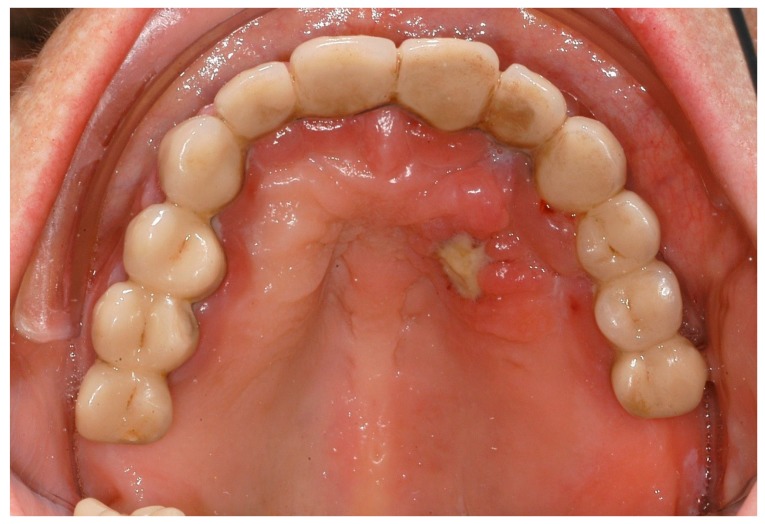
Gingival enlargement and necrosis as a sign of leukaemia in a 75-year-old women.

**Figure 6 ijerph-16-00867-f006:**
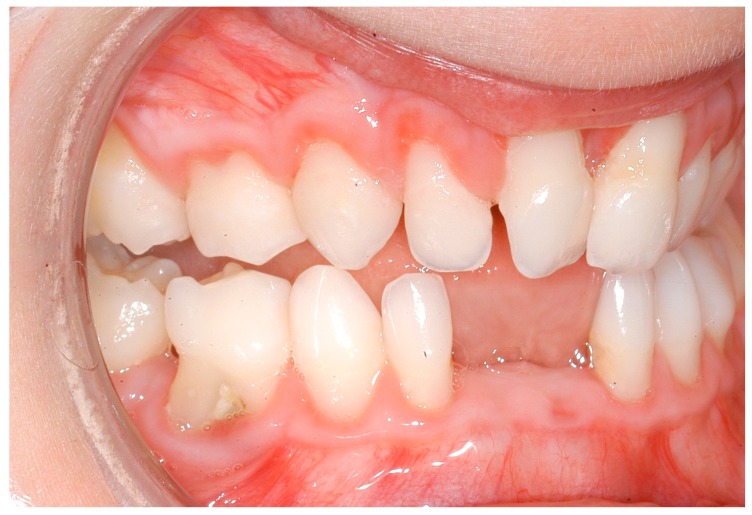
Five-year-old girl with Ehlers–Danlos Syndrome and periodontal destruction.

**Table 1 ijerph-16-00867-t001:** Rare diseases with periodontal manifestations: symptoms, pathological mechanism, affected gene, and classification according the 2018 version of the Classification of Periodontal and Peri-Implant Diseases and Conditions.

Disease Name, ICD-10	Symptoms	Possible Pathological Mechanisms Related with the Periodontal Manifestations	Gene, Location of the Disease	Classification of Periodontal and Peri-Implant Diseases and Conditions
Acrodermatitis enteropathica E83.2	gingivitis	zinc deficiency [40]	*SLC39A4,* *8q24.3* [41]	non-plaque induced gingival diseases and conditions: endocrine, nutritional, and metabolic disease
Acromegaloid facial appearance syndrome Q87.0	gingival enlargement	not reported [42]	*ABCC9*, *exon 27* [41]	non-plaque induced gingival diseases and conditions: genetic/development disorder
Acroosteolysis, dominant type/Hajdu-Cheney Syndrome M89.5	periodontitis [15]	not available	*NOTCH2* *1p12* [41]	systemic diseases and conditions that affect the periodontal attachment apparatus: metabolic and endocrine disorders
Acute megakaryo blastic leukemia C94.2	gingival enlargement gum bleeding	heterogeneous hematologic malignancy characterized by the clonal expansion of myeloid blasts in the peripheral blood, bone marrow, and/or other tissues cause bleeding [43]	not genetically determined	non-plaque induced gingival diseases and conditions: neoplasms
Acute myeloid leukemia C92.0	gingival enlargement gum bleeding	gingival infiltration of leukaemic cells, infiltration of myelomonocytic cells into the bone marrow, destruction of coagulation factors, hypofibrinogenemia [34]	not genetically determined	non-plaque induced gingival diseases and conditions: neoplasms
Acute myelomonocytic leukemia C92.5	gingival enlargement gum bleeding	infiltration of myelomonocytic cells into the gingival tissue [44]	not genetically determined	non-plaque induced gingival diseases and conditions: neoplasms
AL amyloidosis E85.9	gingival enlargement periodontitis	immunoglobulin light chain-derived deposition [35]	not genetically determined	non-plaque induced gingival diseases and conditions: neoplasms
Alopecia-epilepsy-pyorrhea-intellectual disability syndrome Q87.8	gingivitis	unknown [45]	unknown [45]	non-plaque induced gingival diseases and conditions: genetic/developmental disorders
Amelogenesis imperfecta-gingival hyperplasia syndrome K00.5	gingival enlargement	exact pathomechanism unknown [46]	*FAM20A*,*17* *q24.2* [41]	non-plaque induced gingival diseases and conditions: genetic/developmental disorders
Argyria T56.8	gray to gray-black staining of gingiva	deposition of silver and selenium [47]	not genetically determined	non-plaque induced gingival diseases and conditions: gingival pigmentation
Autosomal dominant severe congenital neutropenia D70	periodontitis	neutropenia [48]	*ELANE**19p13.3* [41]	systemic diseases and conditions that affect the periodontal attachment apparatus: diseases associated with immunologic disorders
Bernard Soulier syndrome/Giant platelet syndrome D69.1	gum bleeding bleeding diathesis	low platelet count and macrothrombocytopenia [49]	*GP1BA*,*17p* *13.2* [41]	non-plaque induced gingival diseases and conditions: genetic/developmental disorders
Chediak–Higashi Syndrom E70.3	gingivitis, periodontits	genetic disorder of granule morphology and function [50]	*LYST1q42.3* [41]	systemic diseases and conditions that affect the periodontal attachment apparatus: diseases associated with immunologic disorders
Chronic granulomatous disease D71	periodontitis	defect in metabolic burst for the production of oxygen-free radicals [51]	depending on subtype [41]	systemic diseases and conditions that affect the periodontal attachment apparatus: diseases associated with immunologic disorders
Cohen syndrome Q87.8	periodontitis	neutropenia [33]	*VPS13B*,*8q* *22.2* [41]	systemic diseases and conditions that affect the periodontal attachment apparatus: diseases associated with immunologic disorders
Congenital factor II deficiency D68.2	gum bleeding bleeding diathesis	low prothrombin antigen levels [52]	*F2*,*11p11.2* [41]	non-plaque induced gingival diseases and conditions: genetic/developmental disorders
Congenital factor V deficiency D68.2	gum bleeding bleeding diathesis	low blood coagulation factor V levels [53]	*F5*,*1q24.2* [41]	non-plaque induced gingival diseases and conditions: genetic/developmental disorders
Congenital factor VII deficiency D68.2	gum bleeding bleeding diathesis	low blood coagulation factor VII levels [54]	*F7*,*13q34* [41]	non-plaque induced gingival diseases and conditions: genetic/developmental disorders
Congenital factor X deficiency D68.2	gum bleeding bleeding diathesis	Factor X deficiency [55]	*F10*,*13q34* [41]	non-plaque induced gingival diseases and conditions: genetic/developmental disorders
Congenital factor XI deficiency D68.1	gum bleeding bleeding diathesis	Factor XI deficiency [56]	*F11*,*4q35.2* [41]	non-plaque induced gingival diseases and conditions: genetic/developmental disorders
Congenital factor XII deficiency D68.2	gum bleeding bleeding diathesis	Factor XII deficiency [57]	*F12*,*5q35.3* [41]	non-plaque induced gingival diseases and conditions: genetic/developmental disorders
Congenital factor XIII deficiency D68.2	gum bleeding bleeding diathesis	Factor XIII deficiency [58]	*F13A1*,*6p25.1* *F13B*,*1q31.3* [41]	non-plaque induced gingival diseases and conditions: genetic/developmental disorders
Crohn disease K50	gingival enlargement	chronic inflammation [59]	not genetically determined	non-plaque induced gingival diseases and conditions: granulomatous inflammatory conditions (orofacial granulomatosis)
Cyclic neutropenia D70	mucosal ulcers, periodontitis [60]	fluctuation in the number of blood neutrophils [61]	*ELANE 19q13.3* [41]	systemic diseases and conditions that affect the periodontal attachment apparatus: diseases associated with immunologic disorders
Down syndrome Q90	periodontitis	poor oral hygiene and impairment of the immunological system [36]	*21q22.3* [41]	systemic diseases and conditions that affect the periodontal attachment apparatus: diseases associated with immunologic disorders
Dystrophic epidermolysis bullosa Q81.2	periodontitis	reduced resistance at the junctional epithelial complex [62]	*COL7A1* 3p21.31 [41]	systemic diseases and conditions that affect the periodontal attachment apparatus: diseases affecting the oral mucosa and gingival tissue
Ehlers–Danlos syndrome Q79.6	periodontitis	disturbance of fibrin collagens [63]	*C1R* and *C1S* [63]	systemic diseases and conditions that affect the periodontal attachment apparatus: diseases affecting connective tissues
Familial afibrinogenemi D68.2	gum bleeding bleeding diathesis	fibrinogen deficiency [64]	*FGA, FGB, FGG* [41]	non-plaque induced gingival diseases and conditions: genetic/developmental disorders
Fanconi anemia D61.0	gingivitis	defective hemopoesis [65]	Depending on subtype [41]	non-plaque induced gingival diseases and conditions: genetic/developmental disorders
Focal palmoplantar and gingival keratoderma Q82.8 [66]	gingival hyperkeratosis [66]	not available	Unknown [67]	non-plaque induced gingival diseases and conditions: genetic/developmental disorders
Gaucher disease E75.2 [15]	periodontitis	not available	*GBA**1q22* (Type 1) [41]	systemic diseases and conditions that affect the periodontal attachment apparatus: metabolic and endocrine disorders
Gingival fibromatosis/hypertrichosis syndrome L68.8	gingival enlargement fibromatosis [68]	not available	*17q24.2-q24.3* [41]	non-plaque induced gingival diseases and conditions: genetic/developmental disorders
Gingival fibromatosis-progressive deafness syndrome H90.3	gingival enlargement fibromatosis	unknown [69]	unknown [69]	non-plaque induced gingival diseases and conditions: genetic/developmental disorders
Gingival fibromatosis with facial dysmorphism syndrome Q87.0	gingival enlargement, fibromatosis	unknown [70]	unknown [70]	non-plaque induced gingival diseases and conditions: genetic/developmental disorders
Glanzmann thrombasthenia D69.1	gum bleeding, bleeding diathesis	platelet function disorder [71]	*ITGA2B,**17q21.31* *ITGB3*, *17q21.32* [41]	non-plaque induced gingival diseases and conditions: genetic/developmental disorders
Glycogen storage disease 1b E74.0	periodontal breakdown	neutrophil dysfunction because of endoplasmic retic-ulum stress generated by disruption of endogenous glucose production [33]	*SLC37A4**11q23.3* [41]	systemic diseases and conditions that affect the periodontal attachment apparatus: metabolic and endocrine disorders
Granulomatosis with polyangiitis M31.3	gingival enlargement, periodontitis	production of inflammatory cytocines [13]	not genetically determined	non-plaque induced gingival diseases and conditions: granulomatous inflammatory conditions (orofacial granulomatosis) systemic diseases and conditions that affect the periodontal attachment apparatus: other disorders that may affect periodontal tissue
Hereditary gingival fibromatosis K06.1	gingival enlargement fibromatosis	unknown [72]	*SOS1*, *2p22.1* (Type 1) *5q13-q22* (Type 2) *2p23.3-p22.3* (Type 3) *11p15* (Type 4) *REST*, *4q12* (Type 5) [41]	non-plaque induced gingival diseases and conditions: genetic/developmental disorders
Haim–Munk syndrome/Keratosis palmoplantaris-periodontopathia-onychogry-posis syndrome Q82.8	periodontitis	exact pathomechanism unknown [73]	*CTSC,* *11q14.2* [38,74]	systemic diseases and conditions that affect the periodontal attachment apparatus: diseases associated with immunologic disorders
Hemophilia A D66	gum bleeding bleeding diathesis	Factor VIII deficiency [75]	*F 8*, *Xq28* [41]	non-plaque induced gingival diseases and conditions: genetic/developmental disorders
HemophiliaB D67	gum bleeding bleeding diathesis	Factor IX deficiency [75]	*F 9*, *Xq27.1* [41]	non-plaque induced gingival diseases and conditions: genetic/developmental disorders
Hennekam syndrome/Lymphedema-lymphangiectasia-intellectual disability syndrome Q87.8	gingival enlargement	unknown [76]	*CCBE1*,*18q21.32* (Type 1) *FAT4*,*4q28.* (Type 2) [41]	non-plaque induced gingival diseases and conditions: genetic/developmental disorders
Hereditary angioedema (C-1-inhibitor deficiency) D84.1	periodontitis	deficiency or functional alteration of the C1 inhibitor [77]	*C1NH**11q12.1* (Type 2) *F12 5q35.3* [41]	systemic diseases and conditions that affect the periodontal attachment apparatus: diseases affecting connective tissues
Histoplasmosis D39	gingival candidosis	compromised immunological system [78]	not genetically determined	non-plaque induced gingival diseases and conditions: genetic/developmental disorders
Hyalinosis cutis et mucosae E78.8	gingival enlargement	unknown [79]	*ECM1*, *1q21.2* [41]	non-plaque induced gingival diseases and conditions: genetic/developmental disorders
Hyper-IgE syndrome D82.4	gingivitis, periodontitis [15]	not available	*STAT3 17q21.2; DOCK8* *9q24.3* [41]	systemic diseases and conditions that affect the periodontal attachment apparatus: diseases associated with immunologic disorders
Hyper-IgM syndrome with susceptibility to opportunistic infections D80.5	periodontitis	decreased levels of IgG and IgA; elevated concentrations ofIgM level [80]	*TNFSF5,**Xq26.3* [41]	systemic diseases and conditions that affect the periodontal attachment apparatus: diseases associated with immunologic disorders
Hypertrichosis-acromegaloid facial appearance syndrome Q87.0	gingival enlargement	exact pathomechanism unknown [68]	*17q24.2-q24.3* [41]	non-plaque induced gingival diseases and conditions: genetic/developmental disorders
Hypertrichosis lanuginosa congenita Q84.2	gingival enlargement [81]	not available gingival hyperplasia, is caused by deletion or duplication at chromosome 17q24 or by mutation in the ABCA5 gene	deletion or duplication on *17q2417q24* or mutation in *ABCA5ABCA5* [41,82]	non-plaque induced gingival diseases and conditions: genetic/developmental disorders
Hypophosphatasia E83.38	compromised periodontal attachment	exact pathomechanism unknown [83]	*ALPL**1 p36.12* [41]	systemic diseases and conditions that affect the periodontal attachment apparatus: metabolic and endocrine disorders
Hypophosphatemic rickets E83.30	periodontitis	alteration of bone and tooth mineralization that may lead to malformed and feeble bone and teeth and premature tooth loss [84]	*FGF23* [41]	systemic diseases and conditions that affect the periodontal attachment apparatus: metabolic and endocrine disorders
Hypoplasminogenemia L90.5	Periodontitis, gingival enlargement	exact pathomechanism unknown [85]	*PLG 6q26* [41]	systemic diseases and conditions that affect the periodontal attachment apparatus: diseases affecting the oral mucosa and gingival tissue
Jacobsen syndrome Q93.5	gum bleeding bleeding diathesis	abnormal platelet function and thrombocytopenia [86]	*IC*, *11q23* [41]	non-plaque induced gingival diseases and conditions: genetic/developmental disorders
Juvenile hyaline fibromatosis M72.8	gingival enlargement fibromatosis	unknown [87]	*ANTXR2*, *4q21.21* [41]	non-plaque induced gingival diseases and conditions: genetic/developmental disorders
Juvenile idiopathic arthritis M08.0; M08.1; M08.2; M08.3; M08.4; M08.8; M08.9	periodontal attachment loss [88]	dysregulation of the immune-inflammatory response [88]	not genetically determined	systemic diseases and conditions that affect the periodontal attachment apparatus: diseases associated with immunologic disorders
Kindler syndrome Q81.8	periodontitis	reduced resistance at the junctional epithelial complex [62]	*KIND1 20p12.3* [41]	systemic diseases and conditions that affect the periodontal attachment apparatus: diseases affecting the oral mucosa and gingival tissue
Kostmann syndrome/Severe congenital neutropenia type 3 D70	periodontitis	neutropenia [89]	*HAX1*, *1q21.3* [41]	systemic diseases and conditions that affect the periodontal attachment apparatus: diseases associated with immunologic disorders
Langerhans cell histiocytosis C96	periodontitis	because proliferation of cells with characteristics similar to bone marrow-derived Langerhans cells [15]	not genetically determined	systemic diseases and conditions that affect the periodontal attachment apparatus: other disorders that may affect periodontal tissue
Leukocyte adhesion deficiency type II D84.8	periodontitis	lack of circulating neutrophils due to defect of glycosylation [90]	*SLC35C1*, *11p11.2* [41]	systemic diseases and conditions that affect the periodontal attachment apparatus: diseases associated with immunologic disorders
Majeed syndrome/Chronic recurrent multifocal osteomyelitis-congenital dyserythropoietic anemia-neutrophilic dermatosis syndrome ICD-10: not available	gum bleeding bleeding diathesis	Anemia with dyserythropoiesis [91] Homozygous mutations in LPIN2 Homozygous mutations in LPIN2	*LPIN2*, *18p11.31* [41]	non-plaque induced gingival diseases and conditions: genetic/developmental disorders
Mucolipidosis type II E77.0	gingival enlargement	unknown [92]	*GNPTAB*, *12q23.2* [41]	non-plaque induced gingival diseases and conditions: genetic/developmental disorders
Noonan syndrome Q87.1	gum bleeding bleeding diathesis	bleeding diathesis [93]	Depending on subtype [41]	non-plaque induced gingival diseases and conditions: genetic/developmental disorders
Papillon–Lefèvre syndrome/Keratosis palmoplantar-periodontopathy syndrome Q82.8	periodontitis	exact pathomechanism unknown [72]	*CTSC*, *11q14.2* [41,94]	systemic diseases and conditions that affect the periodontal attachment apparatus: diseases associated with immunologic disorders
Paroxysmal nocturnal hemoglobinuria D59.5	periodontitis	deficiency of monocytic CD14 [95]	*PIGA*, *Xp22.2* (Type 1) *PIGT*, *20q13.12* (Type 2) [41]	systemic diseases and conditions that affect the periodontal attachment apparatus: diseases associated with immunologic disorders
Ramon syndrome/Cherubism-gingival fibromatosis-intellectual disability syndrome Q87.8	gingival enlargement fibromatosis	unknown [96]	depending on subtype, mostly unknown [96]	non-plaque induced gingival diseases and conditions: genetic/developmental disorders
Robinow syndrome/Acral dysostosis with facial and genital abnormalities Q87.1	gingival enlargement	mutant transcript in fibroblasts [97]	*ROR2*, *9q22.31* (Type: autosomal-recessive) *WNT5A*, *3p14.3* (Type: dominant 1) *DVL1,* *1p36.33* (Type: dominant 2) *DVL3*, *3q27.1* (Type: dominant 3) [41]	non-plaque induced gingival diseases and conditions: genetic/developmental disorders
Sarcoidosis D86	gingival enlargement	Cathepsin dysregulation [98]	*HLA-DRB1,**6p21.32* (Type SS1) *BTNL2*, *6p21.32* (Type SS2) *GRCh38*, *10q22.3* (Type SS3) [41]	non-plaque induced gingival diseases and conditions: granulomatous inflammatory conditions (orofacial granulomatosis)
Seckel syndrome Q87.1	gingival enlargement [99]	not available	depending on subtype [41]	non-plaque induced gingival diseases and conditions: genetic/developmental disorders
Segmental odontomaxillary dysplasia K00.4	gingival enlargement	unknown [100]	not genetically determined	non-plaque induced gingival diseases and conditions: genetic/developmental disorders
Severe congenital neutropenia D70	periodontitis	neutropenia [48]	*ELANE*, *19p13.3* [41]	systemic diseases and conditions that affect the periodontal attachment apparatus: diseases associated with immunologic disorders
Systemic lupus erythematosus M32	gingivitis, periodontitis	increased accumulation of immune cells, antineutrophil cytoplasm antibodies and metalloproteinases [101]	*PTPN22**1p13.2* *FCGR2A**1q23.3* *FCGR2B**1q23.3* *CTLA4**2q33.2* *TREX1**3p21.31* *DNASE1**16p13* [41]	systemic diseases and conditions that affect the periodontal attachment apparatus: diseases affecting connective tissues
Systemic sclerosis M34	periodontitis	unknown [102]	not genetically determined	systemic diseases and conditions that affect the periodontal attachment apparatus: other disorders that may affect periodontal tissue
Von Willebrand disease D68.0	gum bleeding bleeding diathesis	deficiency of von Willebrand factor [103]	Depending on subtype [41]	non-plaque induced gingival diseases and conditions: genetic/developmental disorders
Winchester syndrome M89.5	gingival enlargement	mucopolysaccharides stored in tissue [104]	*MMP14,**14q11.2* [41]	non-plaque induced gingival diseases and conditions: genetic/developmental disorders
Wiskott–Aldrich syndrome/Eczema-thrombocytopenia-immunodeficiency syndrome D82.0	gum bleeding bleeding diathesis	thrombozytopenia, neutropenia [105]	*WAS, Xp11.4-p11.21* [41]	non-plaque induced gingival diseases and conditions: genetic/developmental disorders
Zimmerman–Laband syndrome/Gingival fibromatosis-hepatosplenomegaly-other anomalies syndrome Q87.8	gingival enlargement fibromatosis	unknown [106]	*KCNH1*, *1q32.2* (Type 1) *ATP6V1B2*, *8p21.3* (Type 2) [41]	non-plaque induced gingival diseases and conditions: genetic/developmental disorders

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
