# Peer review of "Rare Diseases with Periodontal Manifestations"

_ijerph, 2019, doi:10.3390/ijerph16050867_

Round 1

Reviewer 1 Report

Thank you for your cooperation in this work of great scientific interest.The paper has merit and the topic fits into the Journal aim and scope.

However some  minor concers should be addressed in order to have a final more strong paper. I believe this can be a guideline for the oral surgeon of all public health facilities. The review was well conducted despite some diseases with oral correlations were omitted. There are some chronic inflammatory diseases and some neurodegenerative diseases that deserve to be mentioned.

Author Response

We would like to thank the Editor and the Reviewers for revising our manuscript [ijerph-453119] entitled “Rare Diseases with Periodontal Manifestations“ and the constructive points discussed. The helpful comments and suggestions for improving the manuscript have been incorporated into the revised version and all changes were highlighted “Track Changes”. In this letter, we provide a point-by-point response to each addressed comment and hope the manuscript is now suitable for publication in the International Journal of Environmental Research and Public Health.

Reviewer 1:

Comments and Suggestions for Authors

Thank you for your cooperation in this work of great scientific interest. The paper has merit and the topic fits into the Journal aim and scope.

However some minor should be addressed in order to have a final more strong paper. I believe this can be a guideline for the oral surgeon of all public health facilities. The review was well conducted despite some diseases with oral correlations were omitted. There are some chronic inflammatory diseases and some neurodegenerative diseases that deserve to be mentioned.

We have now added juvenile idiopathic arthritis to the table1. Neurodegenerative diseases such as Parkinson`s and celiac disease have not listed, as they not a rare diseases according to the Orphanet classification of rare diseases in the EU. Thus only rare diseases that are defined as rare in the EU were covered. We have now additionally explained this selected reference in the Methods section (line 76/77).

Reviewer 2:

Comments and Suggestions for Authors

This is a well-written submission, which builds on work already published in IJERPH on this topic, including "Oral Symptoms and Oral Health-Related Quality of Life in People with Rare Diseases in Germany: A Cross-Sectional Study ".

Some comments :

Agree to rely on the Classification of Periodontal and Peri-Implant Diseases and Conditions which introduces "systemic diseases".

It seems to me that references could have been included. For example (unless error):

Molina-García A, Castellanos-Cosano L, Machuca-Portillo G, Posada-de la Paz M. Impact of rare diseases in oral health. Med Oral Patol Oral Cir Bucal. 2016; 21 (5): e587-94. Published 2016 Jul 31. doi: 10.4317 / medoral.20972

A systematic review of the oral and craniofacial manifestations of cri of the cat syndrome. Corcuera-Flores JR, Casttellanos-Cosano L, Torres-Lagares D, Serrera-Figallo MÁ, Rodríguez-Caballero Á, Machuca-Portillo G. Clin Anat. 2016 Jul; 29 (5): 555-60. Epub 2015 Dec 21.

Thank you very much, we have now considered the two publications and discussed them in detail in the discussion section. (line 173-175).

Perhaps one of the limits in finding references to your study:

In Europe, apart from Orphanet, there are rare disease registries (Spain, France, Italy, etc.), as well as reference centers for rare diseases assigned to oral manifestations.

Similarly, your orientation is very European. You do not address, if I am not mistaken, the US context via NIDR, nor the issue of diseases neglected according to the WHO classification that "overlap" with rare diseases (Ex: Noma oro-facial).

In fact, we have focused exclusively on the EU's definition of rare disease and only covered diseases that were listed in the Orphanet list of rare diseases. We started our project after the EU called on politicians to initiate measures to improve the care of people with rare diseases. Therefore, this project was initiated in Europe and is oriented towards the definition of "rare disease" in the EU. Nevertheless, our findings can certainly be transferred to other countries outside the EU (e.g. the USA).

I see proposes to delete the figures relating to clinical cases.

We have consciously tried to underpin our work with clinical examples. If desired by the editor, we can delete the images of course.

Finally, I would find it logical that you submit to a more focused journal in periodontology and / or dentistry for better dissemination

We also see our work as an engagement for the public health system to contribute to the importance of dentistry for the recognition of rare diseases. We therefore consider a public health journal, in particular an open access journal, to be the appropriate publication organ.

Reviewer 3:

Comments and Suggestions for Authors

The work has been well done, and I believe it can become a guideline for the dentistry clinician, during the treatment of special needs patients, that often happens in public health facilities. Congratulations on the high number of diseases taken into consideration. I believe that the study has been carried out correctly, our study team is conducting research on diabetes-related oral diseases at this time. Slightly lengthen the conclusions, speaking of the diagnostic role that oral health can have on these diseases. How, therefore, the dentist can help to make an early diagnosis of pathology, following certain parameters highlighted in this study. Allow me only to suggest some chronic diseases with oral correlations, which have been omitted/missed such as juvenile idiopathic arthritis, neurodegenerative diseases such as parkinson and celiac disease:

Isola, G., Ramaglia, L., Cordasco, G., Lucchese, A., Fiorillo, L., Matarese, G. The effect of a functional appliance in the management of temporomandibular joint disorders in patients with juvenile idiopathic arthritis (2017) Minerva Stomatologica, 66 (1), pp. 1-8.

Cervino, G., Fiorillo, L., Laino, L., Herford, A. S., Lauritano, F., Lo Giudice, G., . . . Cicciù, M. (2018). Oral health impact profile in celiac patients: Analysis of recent findings in a literature review. Gastroenterology Research and Practice, 2018 doi:10.1155/2018/7848735

Fiorillo, L., Cervino, G., Herford, A. S., Lauritano, F., D’Amico, C., Lo Giudice, R., . . . Cicciù, M. (2018). Interferon crevicular fluid profile and correlation with periodontal disease and wound healing: A systemic review of recent data. International Journal of Molecular Sciences, 19(7) doi:10.3390/ijms19071908

Cicciù, M., Risitano, G., Lo Giudice, G., & Bramanti, E. (2012). Periodontal health and caries prevalence evaluation in patients affected by parkinson's disease. Parkinson's Disease, doi:10.1155/2012/541908

We have extended the conclusion somewhat and have discussed the effect of the diseases on oral health and the resulting need for dental treatment and we have also emphasized the role of the dentist in diagnostics (line 221-224).

Thank you very much for your suggestions. We have not listed the celiac disease as well as Parkinson`s disease, as they are not  are diseases according to the Orphanet classification of rare diseases in the EU. Thus only rare diseases that are defined as rare in the EU were covered. We have now additionally explained this in the Methods section (line 76/77).

We have added juvenile idiopathic arthritis to the table 1, thank you very much for your attention!

Thank you for your cooperation, I am happy to contribute to a work of great scientific importance.

Reviewer 4:

Comments and Suggestions for Authors

The concept of this manuscript is interesting and helpful for patients and dentists. However, current form is not sufficient to publish.

 What kind of literature was included for the rationale to decide periodontal symptoms? meta-analysis, systematic review, RCT, or case report? These are completely different regarding the weight of evidence. Authors should show the literature used for determination of those symptoms. It can be informative for readers.

The problem with rare diseases is that by definition they are "rare". Therefore, large case numbers, as required for meaningful RCTs or meta-analyses, are usually not available. Therefore, these are mostly case reports. In the field of rare diseases in connection with dentistry or oral and maxillofacial surgery evidence-based clinical studies and results almost are not present.  We have even deposited the literature describing the pathomechanism for the periodontally relevant symptom in the majority of the listed diseases. In a few of them, there was no available literature on the pathomechanism. In these publications, we have now explicitly deposited literature that supports the described symptoms listed in Table 1. Basing on all of  the above, it can be deduced that preserving the continuation and consequence in spreading the knowledge or rare diseases among dentists and oral and maxillofacial surgeons, as well as patients, is a responsibility of everyone who even once came across this problem. Thus, the current report may add some knowledge and inspire others to go ahead with the development of a database in patients with rare disorders.

-             In page 3, line 105-106, “ If no consensus...expert (TH).” is unclear. Which disease were not reached consensus, and how did author decided? Please describe clearly about this.

TH is professor for "Periodontology". We discussed together whether diseases that occur with insulin-resistant diabetes should automatically be accepted as "periodontally relevant". We decided against it if the actual disease does not cause periodontally relevant symptoms. Thus, only diseases that are clearly associated with periodontal relevance were included.

-             In some disease, the symptoms which indicated in table1 were not sufficient. For example, cyclic neutropenia indicated only “mucosal ulcers”, however, this disease is well known for cause of periodontitis. The literatures described the association between cyclic neutropenia and periodontitis can be found in website such as Pubmed. Therefore, again, literatures used for determination of symptoms need to be shown.

We have even deposited the literature describing the pathomechanism for the parodonally relevant symptom in the majority of the listed diseases. In a few of them, there was no available literature on the pathomechanism. In these publications, we have now explicitly deposited literature that supports the described symptoms in Table 1, see above. For cyclic neutropenia we have now deposited the symptom "periodontits" and inserted a literature reference.

Reviewer 2 Report

This is a well-written submission, which builds on work already published in IJERPH on this topic, including "Oral Symptoms and Oral Health-Related Quality of Life in People with Rare Diseases in Germany: A Cross-Sectional Study ".

Some comments :

Agree to rely on the Classification of Periodontal and Peri-Implant Diseases and Conditions which introduces "systemic diseases".

It seems to me that references could have been included. For example (unless error):

Molina-García A, Castellanos-Cosano L, Machuca-Portillo G, Posada-de la Paz M. Impact of rare diseases in oral health. Med Oral Patol Oral Cir Bucal. 2016; 21 (5): e587-94. Published 2016 Jul 31. doi: 10.4317 / medoral.20972

A systematic review of the oral and craniofacial manifestations of cri of the cat syndrome. Corcuera-Flores JR, Casttellanos-Cosano L, Torres-Lagares D, Serrera-Figallo MÁ, Rodríguez-Caballero Á, Machuca-Portillo G. Clin Anat. 2016 Jul; 29 (5): 555-60. Epub 2015 Dec 21.

Perhaps one of the limits in finding references to your study:

In Europe, apart from Orphanet, there are rare disease registries (Spain, France, Italy, etc.), as well as reference centers for rare diseases assigned to oral manifestations.

Similarly, your orientation is very European. You do not address, if I am not mistaken, the US context via NIDR, nor the issue of diseases neglected according to the WHO classification that "overlap" with rare diseases (Ex: Noma oro-facial).

I see proposes to delete the figures relating to clinical cases.

Finally, I would find it logical that you submit to a more focused journal in periodontology and / or dentistry for better dissemination

Author Response

(The authors gave the same response as above.)

Reviewer 3 Report

The work has been well done, and I believe it can become a guideline for the dentistry clinician, during the treatment of special needs patients, that often happens in public health facilities. Congratulations on the high number of diseases taken into consideration. I believe that the study has been carried out correctly, our study team is conducting research on diabetes-related oral diseases at this time. Slightly lengthen the conclusions, speaking of the diagnostic role that oral health can have on these diseases. How, therefore, the dentist can help to make an early diagnosis of pathology, following certain parameters highlighted in this study. Allow me only to suggest some chronic diseases with oral correlations, which have been omitted/missed such as juvenile idiopathic arthritis, neurodegenerative diseases such as parkinson and celiac disease:

Isola, G., Ramaglia, L., Cordasco, G., Lucchese, A., Fiorillo, L., Matarese, G. The effect of a functional appliance in the management of temporomandibular joint disorders in patients with juvenile idiopathic arthritis (2017) Minerva Stomatologica, 66 (1), pp. 1-8.

Cervino, G., Fiorillo, L., Laino, L., Herford, A. S., Lauritano, F., Lo Giudice, G., . . . Cicciù, M. (2018). Oral health impact profile in celiac patients: Analysis of recent findings in a literature review. Gastroenterology Research and Practice, 2018 doi:10.1155/2018/7848735

Fiorillo, L., Cervino, G., Herford, A. S., Lauritano, F., D’Amico, C., Lo Giudice, R., . . . Cicciù, M. (2018). Interferon crevicular fluid profile and correlation with periodontal disease and wound healing: A systemic review of recent data. International Journal of Molecular Sciences, 19(7) doi:10.3390/ijms19071908

Cicciù, M., Risitano, G., Lo Giudice, G., & Bramanti, E. (2012). Periodontal health and caries prevalence evaluation in patients affected by parkinson's disease. Parkinson's Disease, doi:10.1155/2012/541908

Thank you for your cooperation, I am happy to contribute to a work of great scientific importance.

Author Response

(The authors gave the same response as above.)

Reviewer 4 Report

The concept of this manuscript is interesting and helpful for patients and dentists. However, current form is not sufficient to publish.

-             What kind of literature was included for the rationale to decide periodontal symptoms? meta-analysis, systematic review, RCT, or case report? These are completely different regarding the weight of evidence. Authors should show the literature used for determination of those symptoms. It can be informative for readers

-             In page 3, line 105-106, “ If no consensus...expert (TH).” is unclear. Which disease were not reached consensus, and how did author decided? Please describe clearly about this.

-             In some disease, the symptoms which indicated in table1 were not sufficient. For example, cyclic neutropenia indicated only “mucosal ulcers”, however, this disease is well known for cause of periodontitis. The literatures described the association between cyclic neutropenia and periodontitis can be found in website such as Pubmed. Therefore, again, literatures used for determination of symptoms need to be shown.

Author Response

(The authors gave the same response as above.)

Round 2

Reviewer 3 Report

Thank you for making the required changes, I think now the work is more complete. I also congratulate the reviewer colleagues who helped the authors improve their article.

Regarding the speech of rare diseases I fully agree with the authors. I will send the work in minor revision suggesting only some references, that could improve your work in discussion section. Confident of the good work already done by the authors. Thank you

Best Regards

Cervino, G.; Terranova, A.; Briguglio, F.; ... Fiorillo. L. Diabetes: Oral Health Related Quality of Life and Oral Alterations. Oct 2019, Biomed Research International 

Cicciù, M. Neurodegenerative Disorders and Periodontal Disease: Is There a Logical Connection? Neuroepidemiology. 2016;47(2):94-95. Epub 2016 Sep 13.

Author Response

We would like to thank the Editor and the Reviewers for revising our manuscript [ijerph-453119] entitled “Rare Diseases with Periodontal Manifestations“ and the constructive points discussed. The helpful comments and suggestions for improving the manuscript have been incorporated into the revised version and all changes were highlighted “Track Changes”. In this letter, we provide a point-by-point response to each addressed comment and hope the manuscript is now suitable for publication in the International Journal of Environmental Research and Public Health.

Reviewer 3:

Comments and Suggestions for Authors

Thank you for making the required changes, I think now the work is more complete. I also congratulate the reviewer colleagues who helped the authors improve their article.

Regarding the speech of rare diseases I fully agree with the authors. I will send the work in minor revision suggesting only some references, that could improve your work in discussion section. Confident of the good work already done by the authors. Thank you

Best Regards

Cervino, G.; Terranova, A.; Briguglio, F.; ... Fiorillo. L. Diabetes: Oral Health Related Quality of Life and Oral Alterations. Oct 2019, Biomed Research International 

Unfortunately we couldn't find the article at "Pubmed", "BioMed Research International", “Researchgate” or "Google". Therefore we could not include this article. Do you have the Pubmed-ID?

Cicciù, M. Neurodegenerative Disorders and Periodontal Disease: Is There a Logical Connection? Neuroepidemiology. 2016;47(2):94-95. Epub 2016 Sep 13.

Thank you very much for this important article, we have now considered it in the discussion section (line 196-198).

Reviewer 4:

Comments and Suggestions for Authors

- In page 3, line 105-106, “ If no consensus...expert (TH).” is unclear. Which disease were not reached consensus, and how did author decided? Please describe clearly about this.

This sentence is indeed somewhat unclear and we hope that our methodology is now more understandable. We have tried to establish very precise criteria as to whether a rare disease is considered periodontally relevant or not (see methods section). However, we had agreed in advance that if a dissent prevails, TH will decide whether a disease is periodontally relevant or not. In fact, with the exception of insulin-resistant diabetes, we have been able to include the diseases listed here without debate through our criteria defined in advance.

TH is professor for "Periodontology". We discussed together whether diseases that occur with insulin-resistant diabetes should automatically be accepted as "periodontally relevant". We decided against it if the actual disease does not cause periodontally relevant symptoms. Thus, only diseases that are clearly associated with periodontal relevance were included.

 Please include the underlined sentence in methods section.

Thank you so much for this important addition. We have now inserted this sentence into the methods section (line 106-110).

Reviewer 4 Report

- In page 3, line 105-106, “ If no consensus...expert (TH).” is unclear. Which disease were not reached consensus, and how did author decided? Please describe clearly about this.

TH is professor for "Periodontology". We discussed together whether diseases that occur with insulin-resistant diabetes should automatically be accepted as "periodontally relevant". We decided against it if the actual disease does not cause periodontally relevant symptoms. Thus, only diseases that are clearly associated with periodontal relevance were included.

Please include the underlined sentence in methods section.

Author Response

(The authors gave the same response as above.)
